# ROYAL SOCIETY
# OPEN SCIENCE

statistics/psychology/evolution

computational social science, development, cultural evolution, history

**Author for correspondence:**
Damian J. Ruck
e-mail: druck@utk.edu

# Cultural prerequisites of socioeconomic development

Damian J. Ruck[1,2,3], R. Alexander Bentley[1,2] and Daniel J. Lawson[4,5]

[1]Department of Anthropology, University of Tennessee, 1621 Cumberland Avenue, Knoxville, TN 37996, USA
[2]Center for the Dynamics of Social Complexity, University of Tennessee, 403B Austin Peay, Knoxville, TN 37996, USA
[3]College of Communication and Information, University of Tennessee, 1345 Circle Park Drive, Knoxville, TN 37996, USA
[4]Population Health Science Institute, University of Bristol, Oakfield Grove, Bristol BS8 2BN, UK
[5]Institute of Statistical Sciences, University of Bristol, Fry Building, Woodland Road, Bristol BS8 1TH, UK

DJR, 0000-0001-8678-8852

In the centuries since the enlightenment, the world has seen an increase in socioeconomic development, measured as increased life expectancy, education, economic development and democracy. While the co-occurrence of these features among nations is well documented, little is known about their origins or co-evolution. Here, we compare this growth of prosperity in nations to the historical record of cultural values in the twentieth century, derived from global survey data. We find that two cultural factors, secular-rationality and cosmopolitanism, predict future increases in GDP *per capita*, democratization and secondary education enrollment. The converse is not true, however, which indicates that secular-rationality and cosmopolitanism are among the preconditions for socioeconomic development to emerge.

## 1. Introduction

The last 300 years have brought about increases in health [1], economic development [2], democracy [3] and education [4]. Socioeconomic development has been most pronounced among Western nations and, during the same time period, Western nations have also adopted an unprecedented set of cultural values that make them anomalous in cross-cultural, historical context [5]. American college students, for example, are substantially psychologically different from traditional societies in concepts of fairness, economic decision-making, individualism, independence (versus conformity) and moral reasoning [6].

This unique set of cultural values broadly consist of secular, self-expression, individualism and emancipation values, which are contrasted with religious, survival, collectivist and traditional

values in other nations [5,7–9]. A pertinent question is whether these distinctive cultural values emerged in response to the rising prosperity in Western societies, or, conversely, whether cultural change preceded those developments. In one theory, socioeconomic development must occur before populations shift their priorities to the 'higher-order' cultural values of individualism, self-expression and respect for diversity [10–13]. In an alternative theory, economic development could not occur until ethnocentrism [14] and strong religious customs [15] were supplanted by a value system based on reason and humanism [8,16–18].

This question motivates a study of the time-ordering of socioeconomic development—increases in democratization, economic development, education and life expectancy—and associated cultural values in the twentieth century. Long-run time series for socioeconomic development variables stretch back to the start of the twentieth century and, in some cases, many centuries before that [2,19–21], but detailed surveys of cultural values across the world's nations have only been systematized since 1990, in the form of the World and European Values Survey (WEVS) [22,23]. As discussed below, we extend the cultural values recorded in the WEVS back to the early twentieth century [24] using two observations: firstly, that cultural values form during the first few decades of life [25–28] and second, that these socialized cultural values are resilient through the rest of life, relative to other birth cohorts [24,29].

# 2. Measuring a century of global cultural value change

Starting with the raw World and European values survey (WEVS), we used two exploratory methods in sequence to reduce the 64 common WEVS questions to two orthogonal multi-variate components. First, we use Exploratory Factor Analysis (EFA) to identify nine cultural factors underlying the 476 583 survey responses. We then interpreted each of these cultural factors in terms of a small and unique set of correlated WEVS questions. For example, the factor we label 'Secularism' is highly correlated with WEVS questions such as, 'How important is religion in your life?' and 'How important is God in your life?' (electronic supplementary material). By using EFA in this first step, we create a summary of only the common variance, such that noise, such as measurement error, is reduced.

In the second step, we ran a principal component analysis (PCA) on the EFA-weighted WEVS data from the previous step, which gave us a reduced orthogonal representation of the common WEVS variance. This weighted PCA procedure combines advantageous features of both EFA and PCA. Using PCA reduces colinearity in subsequent regressions. Using EFA makes the components more interpretable by minimizing noise.

The first two principal components (PC) explain 37% of the common WEVS variance. We retained PC1 and PC2 because, in our subsequent multilevel time-lagged regressions, they both show strong linkages to the various measures of socioeconomic development (life expectancy, education, democracy and GDP *per capita*), whereas PC3 and PC4 do not (electronic supplementary material). We label PC1 as secular-rationality and PC2 as cosmopolitanism based on the EFA factors that they are highly Pearson correlated with ($|r| > 0.4$) (table 1).

Secular-rationality is correlated with secularism ($r = 0.76$), political engagement ($r = 0.62$), respect for individual rights ($r = 0.59$) and low prosociality ($r = -0.45$). This means that secular-rational respondents to the WEVS are those who reported, for example, that religion is important in their lives, that they are likely to attend protests or sign petitions, they only pay taxes when coerced and believe that homosexuality and divorce are justifiable. Cosmopolitanism is correlated with trust in out-groups ($r = 0.78$), trust in norm violators ($r = 0.78$) and respect for individual rights ($r = 0.43$). This means cosmopolitan individuals report willingness to have neighbours that are foreign, homosexual, or from another race, as well as believing that homosexuality and divorce are justifiable.

One challenge in our use of the WEVS is that it only stretches back to 1990. To study the slow emergence of socioeconomic development during the twentieth century we extend the time horizon of the WEVS data to 1900 by treating birth decades as representative of historical time periods [24,30]. This is possible because cultural values are formed during the first few decades of life, meaning those who came of age during the 1930s, when surveyed today, will have cultural values that reflect that era. This is based on convergent interdisciplinary evidence from childhood development [28], political belief formation [26,27], prosociality in small-scale societies [25] and neuroscience [31–33].

To treat birth decade as our time variable, we ran additional tests. First, environmental shocks cause transient changes in cultural values at particular time periods [34–37], which could systematically affect

**Table 1.** Cultural factor correlates ($|r| > 0.4$) of the two orthogonal cultural dimensions $R$ and $C$.

| secular-rationality ($R$) | cosmopolitanism ($C$) |
| --- | --- |
| secularism ($r = 0.76$) | trust of out-groups ($r = 0.78$) |
| political engagement ($r = 0.62$) | trust of norm violators ($r = 0.78$) |
| respect for individuals ($r = 0.59$) | respect for individuals ($r = 0.43$) |
| prosociality ($r = -0.45$) | |

certain birth decades more than others. To address this, we use model comparison (twofold cross validation) to show that cultural value differences between birth decades are stable through time [24,29], which tells us that shocks generally affect the entire population, not individual birth decades. Second, in the absence of survey data from the early twentieth century, we use simulations to show that birth decades are representative of past time periods, even in the presence of slow time period effects and uncertainty regarding when a birth decade enters the adult population. We show that the simulation results and the subsequent time-lagged regression results are robust regardless of whether the age of adulthood is assumed to be 0–10, 10–20 or 20–30 years (electronic supplementary material).

# 3. Time ordering socioeconomic development and cultural values

Using Bayesian multilevel time-lagged regressions (Material and methods), we test the detailed time ordering between our two cultural values and four measures of socioeconomic development (table 2). We compare changes in cosmopolitanism $C$ and secular-rationality $R$, with changes in inflation corrected GDP *per capita GDP* [2], democracy $D$ [19], secondary education enrollment $E$ [20] and life expectancy from birth $L$ [21]. As we are using a Bayesian model, we report expected effect sizes and whether 95% credible interval of the posterior distribution excludes zero.

We increase statistical power by including a random effect $\lambda_n$ for each nation $n$, which allows us to include the 109 ten-decade time series for all nations in a single test. To control for the non-independence of nations [38,39], we add a second random effect for language family $\lambda_l$ as a proxy for cultural relatedness [40]. This controls for the possible diffusion of information between culturally similar nations, given that shared cultural and historical origins often facilitate the spread of political and economic changes [41–43].

To test if the results are sensitive to the age when a birth decade begins to influence society, we reran the analysis assuming this age is between 0–10, 10–20 or 20–30 years. Encouragingly, this choice does not qualitatively alter the results. Moreover, our results are also stable when we measure birth decade time series at particular time periods ($p = 1990, 1995, 2000, 2005, 2010$); see the electronic supplementary material.

We confirmed that our analysis is not unduly affected by deriving secular-rationality and cosmopolitanism using our two-stage weighted PCA. First, we substituted secular-rationality with three of its most correlated cultural factors (from the EFA stage): substituting it with secularism first, then with political engagement and finally with respect for individual rights. In all three tests, the key results were unchanged. However, substituting cosmopolitanism with its most correlated factors did change the result: trust in out-groups, trust in norm violators and respect for individual rights all predict GDP better than does democracy (electronic supplementary material).

The regression results in figure 1 show significant effects (greater than 10% with a credible interval of 95% excluding zero) for past cosmopolitanism on future democracy, as well as past secular-rationality on future education and GDP. In addition, we detect the effect of past life expectancy on future democracy and education, as well as past education on future GDP. While time-ordering does not equate to causality, it can rule out causal models: if events of category A consistently lag those of category B, then A did not cause B.

Figure 2 is a directed acyclic graph (DAG) showing how cosmopolitanism and secular-rationality were prerequisites for many indicators of socioeconomic development. The DAG represents the significant effects illustrated in figure 1, such that each edge flows from a source node that predicts a significant future change in the destination node. Both cosmopolitanism and secular-rationality have a positive out-degree (out$_C$ = 1

**Table 2.** Cultural factors used in this study, with number of countries, $N$, represented. Each represents a 10-point time series representing up to the 10 decades of the twentieth century.

| factor | N | data source |
| --- | --- | --- |
| secular-rationality, $R$ | 109 | [22,23] |
| cosmopolitanism, $C$ | 109 | [22,23] |
| GDP *per capita*, GDP | 103 | [2] |
| democracy, $D$ | 101 | [19] |
| secondary education, $E$ | 74 | [20] |
| life expectancy, $L$ | 105 | [21] |

and out$_R$ = 2) and zero in-degree, demonstrating that they are primal in the development sequence. The other primal development variable is life expectancy (out$_L$ = 2 and in$_L$ = 0).

# 4. Origins of secular-rationality and cosmopolitanism

The large global variation in secular-rationality and cosmopolitanism is related to the linguistic and cultural history of nations [44,45]. To quantify this, we make a simple cultural metric that is the sum of the secular-rationality and cosmopolitanism scores, $S_{R+C}$. Figure 3a illustrates the distribution of $S_{R+C}$ on a world map. The regions with the highest $S_{R+C}$ are in Western Europe and their historic colonies in Australasia and the Americas. From inspection, this map suggests that cultural distance from Western Europe in part determines $S_{R+C}$.

Cultural values can be viewed as the 'software of society'. We hypothesize that cultural values can be innovated in one place and spread to another. Evidence suggests that secular-rationality and cosmopolitanism were likely innovated in post-enlightenment Western Europe [16,18]. Under our hypothesis, these cultural values can then diffuse from one nation's population to another, which occurs more readily between nations that are geographically close or linguistically similar because barriers to communication are lower [41,42,46]. Therefore, we expect to see high $S_{R+C}$ in nations that are either linguistically similar or geographically proximate to the Germanic speaking nations in Western Europe.

To test this, figure 3b plots the average $S_{R+C}$ for each language family in our data, which includes the following single language families: Japanese, Korean, Austroasiatic, Tai and Georgian. We classify former-colonial African nations (such as Nigeria) using the Niger-Congo language family, despite many people in the region speaking European languages. We do not classify these nations using European languages because millions of people still speak native languages (full classifications in electronic supplementary material).

As expected, figure 3b shows that European language families are associated with relatively high $S_{R+C}$. High $S_{R+C}$ nations are found in Germanic, Japanese and Uralic (spoken in Hungary, Estonia and Finland) language families. Then with lower, but still positive, $S_{R+C}$ in Italic, Balto-Slavic, Korean and Sino-Tibetan families. On the other hand, the negative $S_{R+C}$ language families are Albanian, Austroasiatic, Tai, Indo-Aryan, Turkic, Semitic, Niger-Congo, Austronesian and Georgian.

Language families are discrete categories, so we use ancestry information as a continuous proxy for cultural proximity to Western Europe. We assume that cultural distance from European nations is correlated with the proportion of modern-day populations descended from historic European populations in 1500 [47]. This is reasonable even if secular-rational and cosmopolitan cultural values are not transmitted directly because migrating Europeans also carry more stable cultural features such as language and religion. Having a similar language and religion reduces barriers between Europe and the new nation, which means secular-rational and cosmopolitan values can diffuse more easily once innovated [41,42,46].

We fit a linear regression to predict the effect of European ancestry on $S_{R+C}$, testing for the influence of the three European linguistic families with large sample sizes ($n > 10$)—Germanic, Italic and Balto-Slavic. As figure 3c shows, Germanic ancestry has the largest effect (slope = 1.53, standard error 0.11 and $p \approx 0$), Italic has a smaller effect (slope = 0.9, with standard error 0.12 and $p \approx 0$) and Balto-Slavic has the smallest effect (slope = 0.58, with standard error 0.1 and $p \approx 0$). Italic languages are culturally closer to Germanic than Balto-Slavic, suggesting cultural distance from Germanic speaking nations is historically important.

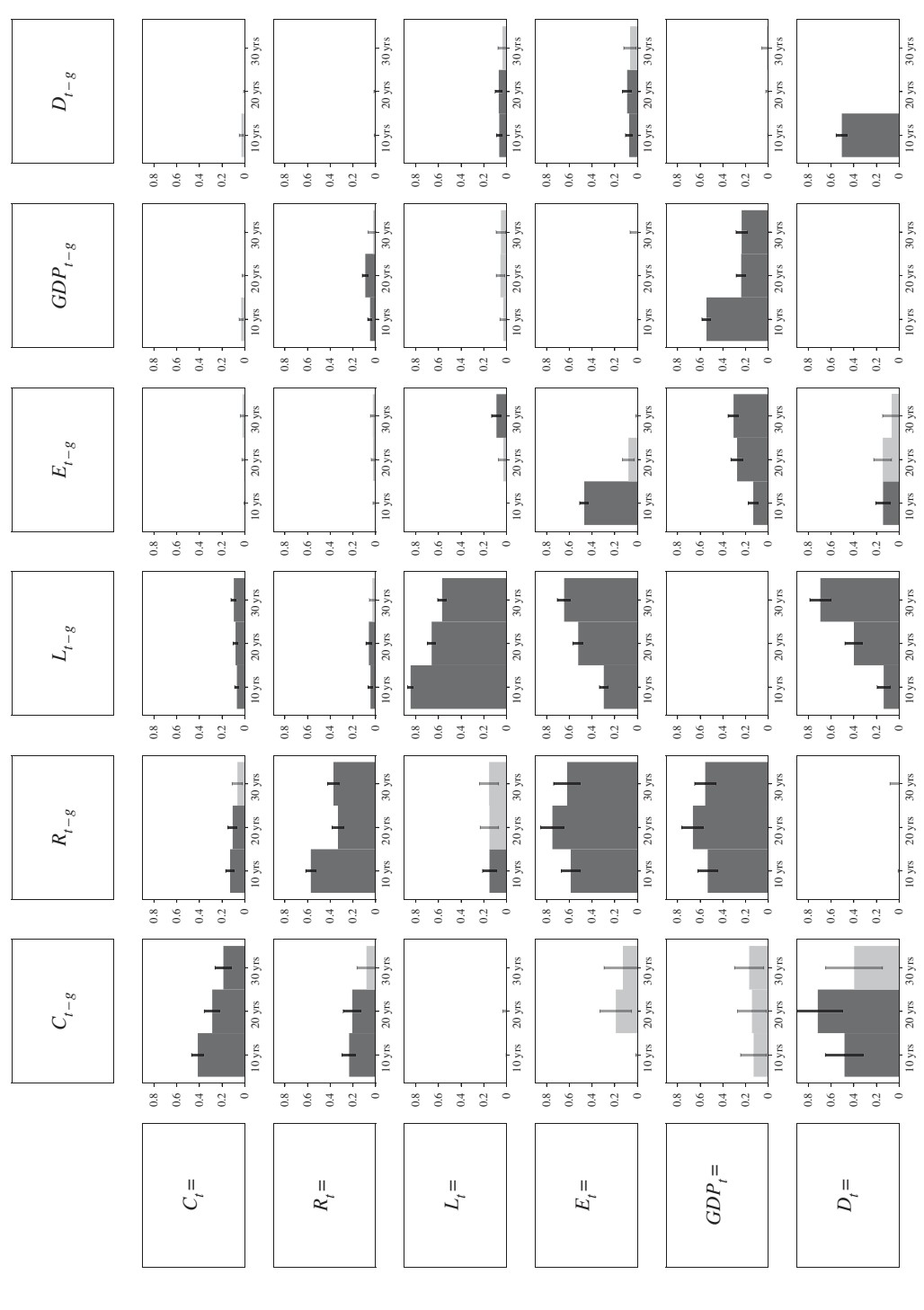

**Figure 1.** Results from the Bayesian multilevel time-lagged linear regression (equation (6.1)). It shows the effect sizes with time lag of $g = 10$, 20 and 20 years for cosmopolitanism (C), secular-rationality (R), life expectancy (L), education (E), GDP *per capita* and democracy (D). If a 95% credible interval excludes zero then the bars are solid and black error bars are standard errors. Full results and diagnostics in electronic supplementary material.

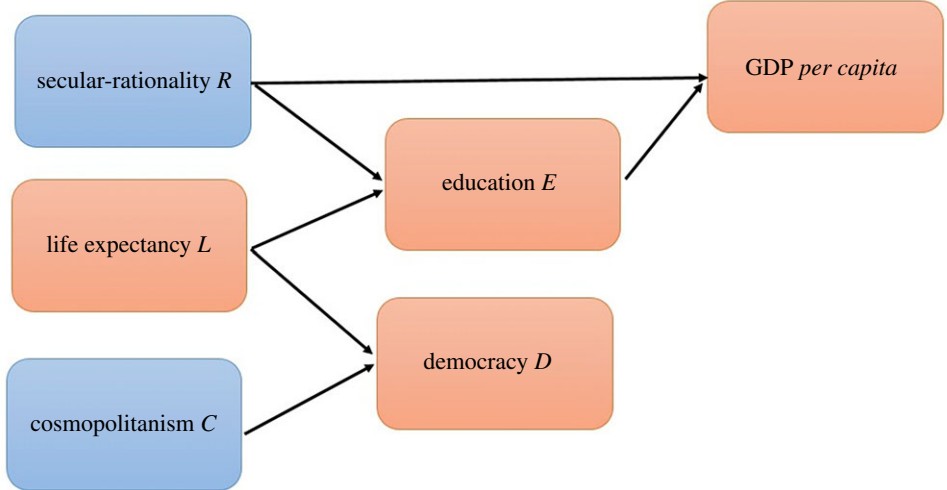

**Figure 2.** A directed acyclic graph (DAG) illustrating the sequence of changes in socioeconomic measures (red boxes) and cultural values (blue boxes). A directed edge means that the variable in the source node has a significant effect on the future value of the destination node.

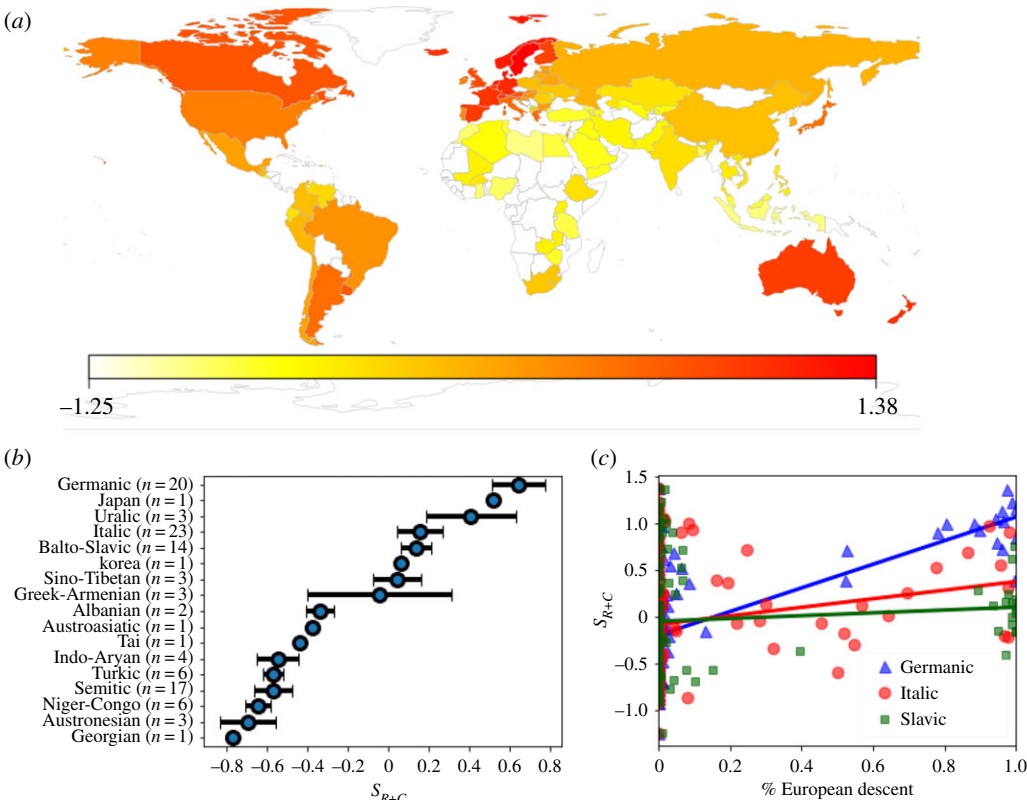

**Figure 3.** (*a*) The 109 WEVS nations on a world map, where nations with a high sum of secular-rationality and cosmopolitanism $S_{R+C}$ are red and those with low $S_{R+C}$ are yellow. (*b*) the mean $S_{R+C}$ for nations in each language family (error bars are standard errors, no error bars for singleton language families). (*c*) Linear fits for $S_{R+C}$ versus % of the population descended from European nations, where blue triangles means descended from Germanic speaking nations, red circles from Italic speaking nations and green squares from Balto-Slavic speaking nations.

## 5. Discussion

This study applied Bayesian multilevel time-lagged linear regressions to investigate the time ordering of various measures of socioeconomic development and cultural values. Time series for secular-rationality and cosmopolitanism were derived from the World and European Values Surveys representing 109

nations over the entire twentieth century [24]. Crucial to extending these time series back to the early twentieth century was the observation that cultural values are crystallized during the first few decades of life [25–28], such that birth decade is a proxy for historical time period [24,30].

As summarized in figure 2, changes in secular-rationality and cosmopolitanism predict future increases in socioeconomic development; where cosmopolitanism leads democracy and secular-rationality leads both economic development and secondary education enrollment. Figure 2 also shows that economic development and democratization are not predictive of future changes, so appear to be the 'end results' of a developmental process involving secular-rationality and cosmopolitanism, in conjunction with higher education enrollment and increased life-expectancy. In a more general sense, we present evidence that secular-rationality and cosmopolitanism were prerequisites for socioeconomic development.

The finding that secular-rationality is primal in the development sequence lends support to certain theories. One theory is that secular-rational values facilitate economic development by focusing human energies to 'use knowledge to enhance human flourishing' [16] rather than practising religious activities [44,48–50]. In this view, the prioritization of secular-technical over spiritual knowledge fuels greater investment in productive activities, which would be consistent with figure 2. However, there appear to be trade-offs. Though secularism predicts greater respect for individual rights and diversity [24], it also predicts a decrease in levels of prosociality (table 2). This supports religious explanations for the evolution of prosocial norms [15].

Cultural values of cosmopolitanism preceded gains in democracy (figure 2). Cosmopolitanism, which includes respect for individuality and equality [5], is likely a prerequisite for functioning democracy. Cosmopolitanism also reflects more tolerance of marginalized groups (table 2)—including homosexuals, drug-users, alcoholics and AIDS sufferers (electronic supplementary material). This is possibly driven by increased exposure [51] to out-groups as the world became more connected during the twentieth century [52,53].

Life expectancy, which has doubled among some Western nations in the past three centuries [54], has had a statistically significant effect on both future democracy and education. The latter is likely explained by the Demographic Transition [55]. In wealthy, knowledge economies, education is a long-term parental investment with payoffs directly related to life expectancy [56–59]. The demographic transition may also explain the link with democracy because older people help in mediating conflicts, whereas young populations will have a greater propensity for violent conflict [60,61]. This 'youth bulge' has been linked with autocracy and democratic recession [62–64].

If, along with the demographic transition, secular-rational and cosmopolitan culture were the drivers of prosperity, how did these values spread? Precisely why secular-rationality and cosmopolitanism arose in Western Europe is not known; it could be due to historic institutional innovations [65,66] or possibly just an accident of history. Nonetheless, once established these values appear to have diffused through geographic proximity—the Uralic speaking nations of Europe have high secular-rationality and cosmopolitanism—and diffused even more readily into nations with similar cultures and languages [41,42,46], giving rise to the current observed pattern of national differences in cultural values (figure 3).

Future work should identify alternative measures of long-term cultural values because the birth decade differences we measure could be distorted by the migration of already culturally socialized individuals [67]. Standard approaches cannot help us correct for this. For example, the use of Putterman and Weil's migration data [47] are not applicable in our case because they do not track the birth decade of migrants. This may not be a concern because migrants do tend to converge on the culture of their new society [68–70], meaning secular-rationality and cosmopolitanism are still likely preconditions for socioeconomic development.

# 6. Methods and data

We used data for the 109 WEVS participating nations. We have annual data for *GDP, D, L* and *E*, but we took decadal averages to correspond with the WEVS birth decade time series.

## 6.1. World and European values survey

We measured cultural values using the world values survey (WVS) [22] and European values survey (EVS) [23], which combined have administered the same 64 questions over a 25-year period to 476 583

participants from 109 unique nations. The surveys were administered in five waves at 5-year intervals, beginning in 1990. Not all nations were available for each wave of the survey, but still 84/109 were asked the same 64 (ordinal scale) questions more than once. Missingness was limited (1.6%), so mean imputation was adequate.

## 6.2. Economic development

We used historical data on GDP *per capita* (in 1990 US$), for the entire twentieth century (1900–2000), provided by the Maddison Project. We have no data for 6/109 of the WEVS nations (Northern Ireland, Malta, Luxembourg, Iceland, Andorra and Cyprus), so we have 10-point time series for 103 nations. We have partial time series for some nations in sub-Saharan Africa (e.g. Nigeria and Burkina Faso) and former Soviet states (e.g. Ukraine, Belarus, Russia). For historical continuity, the following nations are considered the same: Cape Colony has been equated with South Africa; Holland with The Netherlands; Eritrea with Ethiopia, North and Central Italy with Italy, and Great Britain and England with the UK. See [2] for more details.

## 6.3. Democracy

Democracy scores are taken from the Polity IV project [19]. Each nation is assigned a democracy and autocracy score based on features of its electoral process and chief executive restrictions. The final polity score is democracy minus autocracy because a nation can simultaneously have both democratic and autocratic features. Only 8/109 were completely missing (Andorra, Turkish Cyprus, Hong Kong, Iceland, Malta, North Ireland, Palestine and Puerto Rico). See [19] for more details.

## 6.4. Secondary education enrollment

Secondary education enrollment time series between 1900 and 2000 were provided by the 'Barro-Lee Educational Attainment' project, using data taken mainly from censuses and intergovernmental organizations [20]. However, we only have data for 74/109 WEVS nations because small nations and semi-autonomous regions are not included (Northern Ireland, for example), some nations have only recently become independent states (former Soviet and Yugoslav nations) and many poor nations have not collected data (certain African nations, for example).

## 6.5. Life expectancy

Twentieth century time series for life expectancy were compiled by the Clio-infra Project from sources including the United Nations, Human Morality Database, GAPMINDER, the Organization for Economic Co-operation and Development (OECD) and national sources [21]. We have some time-series data for 105/109 WEVS nations (data are missing for: Andorra, Turkish Cyprus, Kosovo and Northern Ireland).

## 6.6. Language family

We used linguistic history $l$ as a proxy for cultural relatedness for use in the random effect ($\lambda_l$). Data were taken from the Ethnologue database [71] containing all extant languages, their phylogenies and the nations where they are spoken. To form discrete categories from continuous trees, we chose split points that gave us reasonable sample sizes. The families were: Albanian, Semitic, Italic, Greek-Armenian, Germanic, Turkic, Indo-Aryan, Balto-Slavic, Sino-Tibetan, Uralic, Georgian, Austronesian, Japanese, Niger-Congo, Korean, Tai and Austroasiatic. See electronic supplementary material for nations categorizations.

## 6.7. European ancestry

Using data from 38 sources, Louis Putterman and David Weil estimated the ancestral nations (in 1500) of modern-day national populations [47], accounting for stateless ethnic groups and modern day mixed ethnic groups. Data included only sovereign nations with a population of greater than 500 000, so Andorra, Hong Kong, Palestine, Kosovo and Northern Ireland were not included.

## 6.8. Using birth decade as a proxy for historical time periods

The WEVS has only been comprehensively carried out since 1990 and this is not long enough to capture the slow intergenerational dynamics that link to socioeconomic development [24,29]. However, convergent interdisciplinary evidence [25–28] suggests that decade of birth can be used to represent historical time periods.

We break down each of the five WEVS survey periods $p$ by birth decade $t$, which yields a matrix of $X_{t,p}$ for each country (for inclusion, the sample for each birth decade must contain at least 100 individuals). We showed that birth decade trends are approximately independent of time period (electronic supplementary material) which tells us that birth decade differences in cultural values persist through time. Before averaging birth decades over all time periods, we account for missing birth decades that are not represented in all time periods. We impute using best-fit birth decade trends to avoid bias stemming from period effects. We fitted birth decade trends using both a linear and quadratic model for robustness:

$$X_{t,p} = \mu_p + \alpha_p t$$

and

$$X_{t,p} = \mu_p + \alpha_p t + \gamma_p t^2$$

where $t$ is the birth decade, $p$ is the time period, $\mu_p$ are the intercepts, $\alpha_p$ is the linear coefficient and $\gamma_p$ is the quadratic coefficient. Our results are stable for both choices of imputation (electronic supplementary material).

Once missing values were imputed, we defined the birth decade time series $X_t$ by averaging across all survey periods $p$. This gives us up to a 10 point time series for the 109 countries in the WEVS (some countries have only partially complete time series if certain birth decades were not surveyed at any time period $p$). Though not strictly equal to past cultural values, birth decade time-series capture long-run cultural value dynamics (electronic supplementary material).

## 6.9. Bayesian multi-level time-lagged regression

We use the following Bayesian time-lagged multilevel linear regressions to detect the sequence of democracy $D$, economic development $GDP$, life-expectancy $L$, secondary education enrollment $E$, cosmopolitanism $C$ and secular-rationality $R$:

$$
\left.
\begin{aligned}
Y_{t,i} &\sim \text{Normal}\,(\nu, \epsilon) \\
\nu_t &= \mu + \lambda_l + \lambda_n + \sum_{j=1}^{J} \beta_{i,j} X_{t-g,j} \\
\mu &\sim \text{Normal}\,(0, 10) \\
\lambda_l &\sim \text{Normal}\,(0, 10) \\
\gamma_n &\sim \text{Normal}\,(0, 10) \\
\beta_{i,j} &\sim \text{Normal}\,(0, 1) \\
\epsilon &\sim \text{HalfCauchy}\,(0, 2)
\end{aligned}
\right\}
\tag{6.1}
$$

where $\beta_{i,j}$ is the effect size of independent variable $j$ in matrix $X_{t-g,j}$ on dependent variable $i$ in matrix $Y_{t,i}$, $\epsilon$ is the error and $\mu$, $\lambda_l$ and $\lambda_n$ are the respective global, language category and nation intercepts. The weakly informative priors capture the fact that all variables are z-scored, which makes large parameter values very unlikely.

The results in figure 1 are produced using the above model when $Y_{t,i}$ and $X_{t,i}$ are as follows:

$$Y_{t,i} = [\,R, C, GDP, D, E, L\,]_t$$

and

$$X_{t-g,j} = [\,R, C, GDP, D, E, L\,]_{t-g}$$

Data accessibility. Data and relevant code for this research work are stored in GitHub: https://github.com/damianruck/Cultural-prerequistes-for-socioeconomic-development and have been archived within the Zenodo repository: https://doi.org/10.5281/zenodo.3559789.

Authors' contributions. All authors contributed equally to data analysis and writing and give final approval for publication.
Competing interests. The authors declare that they have no competing financial interests.
Funding. D.J.R. is funded by the College of Arts and Sciences, University of Tennessee. D.J.L is funded by the Wellcome Trust and Royal Society Sir Henry Dale Fellowship, grant no. WT104125MA.
Acknowledgments. Funding for open access to this research was provided by University of Tennessee's Open Publishing Support Fund.

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
