## [Reviewer comments · Royal Society Open Science]

Review History

RSOS-190725.R0 (Original submission)

Review form: Reviewer 1

Is the manuscript scientifically sound in its present form?

Yes

Are the interpretations and conclusions justified by the results?

No

Is the language acceptable?

Yes

Do you have any ethical concerns with this paper?

No

Have you any concerns about statistical analyses in this paper?

No

Recommendation?

Major revision is needed (please make suggestions in comments)

Comments to the Author(s)

This paper measures economic growth and WEIRD values over time, finding that secular-rationalism and cosmopolitanism precede economic growth, but not vice versa. The authors conclude that these value are precedents of WEIRD culture. I was intrigued by the analysis and the questions being asked, but I see two major issues:

1. I'm not sure I think the conclusion is warranted. As I outline in more detail in Point #1 below, I think Henrich and colleagues would say these values ARE WEIRDness, not pre-requisites. If I'm correct in this, then the paper needs to be entirely re-framed.

2. I was confused throughout the paper about the data and the analysis (points 5, 6, 7, 8 below). I walked away mostly understanding what the conclusion was but without a clear sense of having "seen" the data. I think it is possible to give readers a better sense of the data. For example, the Grossmann and Varnum 2015 paper walks readers through the process of testing time series data quite clearly. They give graphs that explain what's going on. Although Grossmann and Varnum used complicated methods, I walked away having a sense that I had "seen" the data. For this paper, I really only feel like I have the conclusion the authors. I can't exactly say how to fix this, but I think it's worth time on the authors' part to try to improve.

I outline major and minor points below.

Major Points

1. I'm not sure what it means to say that secular-rationality preceded WEIRDness. Isn't secular-rationality WEIRDness itself? Yes, the WEIRD acronym refers to things like education and economic development, but Henrich and colleagues describe these as *correlates* of cultural WEIRDness. What they're primarily interested in is the WEIRD cultural traits of things like trust and analytic thought. Thus, I worry the paper has the logic backwards here. I'm open to being convinced otherwise.

2. For a paper that takes the idea of cultural change so seriously, I was surprised to see the following speculative claim of happenstance, couched in such certain terms:

"Through an accident of history, Secular-Rationality, Cosmopolitanism arose as founding principles of western European nations..."

Huh? How do we know that's a happenstance and not a change with a meaningful cause? It seems far too speculative to claim this.

3. The paper mentions the limitation of needing to look at migration, but doesn't seem to do much about it. It seems to me like two strategies are available: (1) statistically model movement between nations (following the Putterman and Weil paper) and (2) analyze heritage/mobility/birthplace data in the WVS itself (if it exists).

4. Page 2 says, "Long-run time series for socio-economic development variables stretch back to the 19th Century..." I see the authors cite the Maddison project, so perhaps they know this already, but the project includes data (for certain countries) far earlier than the 19th century.

5. Page 3 lists what appear to be either correlations or components of the scales, but it's unclear to me: "is secular ($r = 0.76$), politically engaged ($r = 0.62$)..." Please make this clear. The original scale wordings (at least a few) would give readers a lot better idea of what they're reading.

6. Page 4 describes the use of birth decade and gives the impression the authors did commendable work demonstrating that this is a robust way of analyzing the data. It sounded impressive to me, but I didn't really understand it! One of my biggest questions was, did you try modeling it other ways? For example, what about the possibility that people's values are formed

mostly from age 10-20? Perhaps this is already in the analysis. But what I'm asking is the authors to make these statements clear enough that a casual (but educated) reader can understand concretely what the analyses were. Sometimes communicating analyses clearly is harder than doing the analyses themselves!

UPDATE: On page 5, the paper seems to address this question. That's great! But the fact that it's addressed on page 5 makes me even more confused about what I had read on page 4. What did I read??

LATER UPDATE: The methods say "However convergent interdisciplinary evidence (6; 27; 28; 29) suggests that decade of birth can be used to represent historical time dynamics that link to socioeconomic development (20; 32)."

That's great that there's evidence to support this! But can you describe some of it? Even just a few sentences communicating what those studies have compared to test this idea would be helpful.

7. Page 4 describes the analysis using an alphabet soup that is cognitively taxing for the readers: "changes in C and R...past L on future D." Yes, these are defined elsewhere, so it's possible for readers to figure it out. But why make their job so hard? It adds too much complexity for the readers to try to keep all these symbols in mind. Replacing a single letter with a word or two (like "democracy") will be well worth the extra space taken up. It's far worth the cost in order to have readers be able to actually understand the paper.

8. Page 6 mentions "Uralic" languages. My guess is even most educated readers won't know what that means. It would be helpful to describe that (even just parenthetically) in the paper. (Ditto for Kartvelian!)

Minor Points

1. For this sentence, I think readers would appreciate a basic correlation to get an idea of the relationship. "These two components have strong linkages to measures of socioeconomic development (Supplementary Materials)."
2. The paper says the analysis has a "credible interval of 95%." Is that another term for confidence interval? I've never heard this term before.
3. Page 10: "how that" sounds like a typo to me.
4. Page 8: "the later" should be "the latter."
5. Japanese and Kartvelian are described as language families, yet both of these represent a single language. In those cases, the wording in the paper will mislead readers who don't already know this (somewhat obscure) fact.

Review form: Reviewer 2 (Chico Quevedo Camargo)

Is the manuscript scientifically sound in its present form?

Yes

Are the interpretations and conclusions justified by the results?

Yes

Is the language acceptable?

Yes

Do you have any ethical concerns with this paper?

No

Have you any concerns about statistical analyses in this paper?

No

Recommendation?

Accept with minor revision (please list in comments)

Comments to the Author(s)

This is a very interesting paper. In my opinion, it only needs some minor changes.

I appreciate that the authors put their data and code on github. But as it is right now, there is no sort of README file or guidance on how to reproduce their data. To replicate, I would simply try to run the two source codes they provide, but I would really appreciate some file telling me what to run to produce what figures.

One bit that concerned me is the use of language similarity as a proxy for cultural proximity to western Europe. To me, this is concerning because while many former colonies do share the language of their colonisers, it does not imply at all that they share their values at the time, or even today. While the authors do point that out on page 11 ("This is reasonable because migrating Europeans carried their culture with them, which does not necessarily mean transplanting historic cultural values of the day, but just more stable cultural features like language"), I really do not see why sharing a language would imply sharing values.

To take the English language as an example: the UK, Belize, Guyana, Jamaica and the USA speak English. And as shown in the Supplementary Information, the Germanic family includes Austria, Germany and the UK, but also Singapore, Zambia, Ghana and Nigeria. Would they all have a regression term corresponding to the English language? Perhaps I understood this wrong, but this sounds concerning to me. Perhaps this is addressed by the references (35; 36; 43) cited in the end of the first paragraph of page 11, but as the references are not numbered I am not sure which reference is which. In any case: I think the authors should discuss this assumption -- and this classification of such a wide variety of countries into "Germanic countries", for instance -- a bit further.

Other than that: on page 9, the authors say "Uralic speaking countries of Europe exhibit high S^{R+O} ". Would that be S^{R+C} ?

Decision letter (RSOS-190725.R0)

30-Oct-2019

Dear Dr Ruck,

The editors assigned to your paper ("Cultural prerequisites for WEIRD societies") have now received comments from reviewers. We would like you to revise your paper in accordance with the referee and Associate Editor suggestions which can be found below (not including confidential reports to the Editor). Please note this decision does not guarantee eventual acceptance.

Please submit a copy of your revised paper before 22-Nov-2019. Please note that the revision deadline will expire at 00.00am on this date. If we do not hear from you within this time then it will be assumed that the paper has been withdrawn. In exceptional circumstances, extensions may be possible if agreed with the Editorial Office in advance. We do not allow multiple rounds of revision so we urge you to make every effort to fully address all of the comments at this stage. If deemed necessary by the Editors, your manuscript will be sent back to one or more of the

original reviewers for assessment. If the original reviewers are not available, we may invite new reviewers.

- Data accessibility

If you wish to submit your supporting data or code to Dryad (<http://datadryad.org/>), or modify your current submission to dryad, please use the following link:
<http://datadryad.org/submit?journalID=RSOS&manu=RSOS-190725>

- Competing interests

- Authors' contributions

AB carried out the molecular lab work, participated in data analysis, carried out sequence alignments, participated in the design of the study and drafted the manuscript; CD carried out

the statistical analyses; EF collected field data; GH conceived of the study, designed the study, coordinated the study and helped draft the manuscript. All authors gave final approval for publication.

- Acknowledgements

- Funding statement

Kind regards,
Anita Kristiansen
Editorial Coordinator
Royal Society Open Science
openscience@royalsociety.org

on behalf of Professor Len Thomas (Associate Editor) and Mark Chaplain (Subject Editor)
openscience@royalsociety.org

Associate Editor's comments (Professor Len Thomas):

Comments to the Author:

Your paper has been read by two reviewers. The first raises a number of issues, including concern about the logic underlying the main premise that secular-rationality preceded WIERDness. They also found your analyses hard to follow from the descriptions in the main paper. The second reviewer was more positive, raising only minor concerns. Given this, I am recommending major revision of the paper. Please address the reviewers' concerns either by revising the paper itself or, if you disagree, by responding fully in your response letter. Regarding the analysis description, I would like to see more detail on this in the main paper, rather than in supplemental materials.

In addition to the reviewers' comments, I would also like the following points addressed.

The regression model appears to be Bayesian; this should be mentioned in the main text, together with an explanation for the priors used. I also wondered why a Bayesian analysis was used here when the previous analyses (PCA, etc) are not Bayesian.

I wondered why three different information-criteria were used for inference? I also note that, unlike what is stated in the text, they give different results. From supp materials Fig 4, DIC and WAIC are lower for model M3 compared with model M2.2, certainly for the Cosmopolitanism value. It's hard to tell exactly how much lower from the figure, but it seems to be >50, which is a lot in IC terms.

Thank-you for providing code and data with the paper in a github repository. Unfortunately, it is currently not in an acceptable state for publication with the paper. Please, at a minimum: (i) add a readme file explaining what the files are and how to run them; (ii) add comments both to the R and python files; (iii) tidy both files up so that they use consistent vertical spacing (the code currently is not organized into sensible "paragraphs" and seems to have random single, double and triple vertical spaces); (iv) put all data files into either the top directory or a data directory that is a subdirectory of the top directory (one is currently in the top directory, another is in a directory one level down and the others are a subdirectory nested 4 below the top directory), (v) ensure that the code contains everything needed to replicate the analyses in the paper *and* the

supplemental materials. Please double check whether any more tidying up may be helpful in enabling code reuse.

Minor editorial comments:

Page 4: I could not understand the “five different waves p” in the following sentence “We extended the reach of WEVS surveys, conducted five different waves p over the past 30 years, back to the early 20th century by treating the cultural values in survey responses as representative of the respondent’s birth decade, t (20; 26).” For one thing the p should be enclosed in commas as it is parenthetical. But secondly I’m not sure what these “waves” are. (This is explained in the methods but should be defined where first used.)

Page 11, please define the +- notation used after the estimated slopes in the linear regression results. Are these standard errors? If so, perhaps just say this and remove the +- device.

Page 14, In equation 1, should the β_j in the second formula from bottom be β_{ij} ? Also, unless I missed it, J is not defined.

Reviewers' Comments to Author:

Reviewer: 1

Comments to the Author(s)

This paper measures economic growth and WEIRD values over time, finding that secular-rationalism and cosmopolitanism precede economic growth, but not vice versa. The authors conclude that these value are precedents of WEIRD culture. I was intrigued by the analysis and the questions being asked, but I see two major issues:

1. I'm not sure I think the conclusion is warranted. As I outline in more detail in Point #1 below, I think Henrich and colleagues would say these values ARE WEIRDness, not pre-requisites. If I'm correct in this, then the paper needs to be entirely re-framed.
2. I was confused throughout the paper about the data and the analysis (points 5, 6, 7, 8 below). I walked away mostly understanding what the conclusion was but without a clear sense of having "seen" the data. I think it is possible to give readers a better sense of the data. For example, the Grossmann and Varnum 2015 paper walks readers through the process of testing time series data quite clearly. They give graphs that explain what's going on. Although Grossmann and Varnum used complicated methods, I walked away having a sense that I had "seen" the data. For this paper, I really only feel like I have the conclusion the authors. I can't exactly say how to fix this, but I think it's worth time on the authors' part to try to improve.

I outline major and minor points below.

Major Points

1. I'm not sure what it means to say that secular-rationality preceded WEIRDness. Isn't secular-rationality WEIRDness itself? Yes, the WEIRD acronym refers to things like education and economic development, but Henrich and colleagues describe these as *correlates* of cultural WEIRDness. What they're primarily interested in is the WEIRD cultural traits of things like trust and analytic thought. Thus, I worry the paper has the logic backwards here. I'm open to being convinced otherwise.
2. For a paper that takes the idea of cultural change so seriously, I was surprised to see the following speculative claim of happenstance, couched in such certain terms:

"Through an accident of history, Secular-Rationality, Cosmopolitanism arose as founding principles of western European nations..."

Huh? How do we know that's a happenstance and not a change with a meaningful cause? It seems far too speculative to claim this.

3. The paper mentions the limitation of needing to look at migration, but doesn't seem to do much about it. It seems to me like two strategies are available: (1) statistically model movement between nations (following the Putterman and Weil paper) and (2) analyze heritage/mobility/birthplace data in the WVS itself (if it exists).

4. Page 2 says, "Long-run time series for socio-economic development variables stretch back to the 19th Century..." I see the authors cite the Maddison project, so perhaps they know this already, but the project includes data (for certain countries) far earlier than the 19th century.

5. Page 3 lists what appear to be either correlations or components of the scales, but it's unclear to me: "is secular ($r = 0.76$), politically engaged ($r = 0.62$)..." Please make this clear. The original scale wordings (at least a few) would give readers a lot better idea of what they're reading.

6. Page 4 describes the use of birth decade and gives the impression the authors did commendable work demonstrating that this is a robust way of analyzing the data. It sounded impressive to me, but I didn't really understand it! One of my biggest questions was, did you try modeling it other ways? For example, what about the possibility that people's values are formed mostly from age 10-20? Perhaps this is already in the analysis. But what I'm asking is the authors to make these statements clear enough that a casual (but educated) reader can understand concretely what the analyses were. Sometimes communicating analyses clearly is harder than doing the analyses themselves!

UPDATE: On page 5, the paper seems to address this question. That's great! But the fact that it's addressed on page 5 makes me even more confused about what I had read on page 4. What did I read??

LATER UPDATE: The methods say "However convergent interdisciplinary evidence (6; 27; 28; 29) suggests that decade of birth can be used to represent historical time dynamics that link to socioeconomic development (20; 32)."

That's great that there's evidence to support this! But can you describe some of it? Even just a few sentences communicating what those studies have compared to test this idea would be helpful.

7. Page 4 describes the analysis using an alphabet soup that is cognitively taxing for the readers: "changes in C and R...past L on future D." Yes, these are defined elsewhere, so it's possible for readers to figure it out. But why make their job so hard? It adds too much complexity for the readers to try to keep all these symbols in mind. Replacing a single letter with a word or two (like "democracy") will be well worth the extra space taken up. It's far worth the cost in order to have readers be able to actually understand the paper.

8. Page 6 mentions "Uralic" languages. My guess is even most educated readers won't know what that means. It would be helpful to describe that (even just parenthetically) in the paper. (Ditto for Kartvelian!)

Minor Points

1. For this sentence, I think readers would appreciate a basic correlation to get an idea of the relationship. "These two components have strong linkages to measures of socioeconomic development (Supplementary Materials)."

2. The paper says the analysis has a "credible interval of 95%." Is that another term for confidence interval? I've never heard this term before.

3. Page 10: "how that" sounds like a typo to me.

4. Page 8: "the later" should be "the latter."

5. Japanese and Kartvelian are described as language families, yet both of these represent a single language. In those cases, the wording in the paper will mislead readers who don't already know this (somewhat obscure) fact.

Reviewer: 2

Comments to the Author(s)

This is a very interesting paper. In my opinion, it only needs some minor changes.

I appreciate that the authors put their data and code on github. But as it is right now, there is no sort of README file or guidance on how to reproduce their data. To replicate, I would simply try to run the two source codes they provide, but I would really appreciate some file telling me what to run to produce what figures.

One bit that concerned me is the use of language similarity as a proxy for cultural proximity to western Europe. To me, this is concerning because while many former colonies do share the language of their colonisers, it does not imply at all that they share their values at the time, or even today. While the authors do point that out on page 11 ("This is reasonable because migrating Europeans carried their culture with them, which does not necessarily mean transplanting historic cultural values of the day, but just more stable cultural features like language"), I really do not see why sharing a language would imply sharing values.

To take the English language as an example: the UK, Belize, Guyana, Jamaica and the USA speak English. And as shown in the Supplementary Information, the Germanic family includes Austria, Germany and the UK, but also Singapore, Zambia, Ghana and Nigeria. Would they all have a regression term corresponding to the English language? Perhaps I understood this wrong, but this sounds concerning to me. Perhaps this is addressed by the references (35; 36; 43) cited in the end of the first paragraph of page 11, but as the references are not numbered I am not sure which reference is which. In any case: I think the authors should discuss this assumption -- and this classification of such a wide variety of countries into "Germanic countries", for instance -- a bit further.

Other than that: on page 9, the authors say "Uralic speaking countries of Europe exhibit high S^{R+O} ". Would that be S^{R+C} ?

Author's Response to Decision Letter for (RSOS-190725.R0)

See Appendix A.

Decision letter (RSOS-190725.R1)

16-Dec-2019

Dear Dr Ruck,

It is a pleasure to accept your manuscript entitled "Cultural prerequisites for socioeconomic development" in its current form for publication in Royal Society Open Science. The comments of the reviewer(s) who reviewed your manuscript are included at the foot of this letter.

Please ensure that you send to the editorial office an editable version of your accepted

manuscript, and individual files for each figure and table included in your manuscript. You can send these in a zip folder if more convenient. Failure to provide these files may delay the processing of your proof. You may disregard this request if you have already provided these files to the editorial office.

on behalf of Professor Len Thomas (Associate Editor) and Mark Chaplain (Subject Editor)
openscience@royalsociety.org

Associate Editor Comments to Author:

Thank-you for providing such a detailed response, and for making changes in line with reviewer and my suggestions. I'm happy to recommend the paper for publication.

(I note that the code is still a bit messy, with variable line spacing, few internal comments, commented out lines, etc - however with the provided readme file it is possible to navigate it so OK to accept.)

Appendix A

Manuscript Number: RSOS-190725

Title: Cultural prerequisites for WEIRD societies (revised to “Cultural prerequisites of socioeconomic development”)

Authors: Damian J. Ruck, R. Alexander Bentley and Daniel J. Lawson

November 9, 2019

Professor Len Thomas

Associate Editor, *Royal Society Open Science*

Dear Professor Thomas,

We are pleased that you are considering our manuscript for publication in *Royal Society Open Science*. In response to reviewer one, we have renamed our manuscript “Cultural prerequisites of socioeconomic development”.

We take confidence from the encouraging comments from the reviewers. Reviewer two believes the manuscript only requires very minor changes. Though reviewer one wanted to see some additional revisions, we found resolving these have improved the quality of the paper.

Attached to this letter is a rebuttal to each of the reviewer's points taken in turn. In the rebuttal, the reviewers' comments are in *black italics*, **our responses are in red**, and **changes to manuscript are in blue**.

We have uploaded the point-by-point responses, revised manuscript and supplementary materials to the Royal Society web portal. In addition, we have included statements for data accessibility, competing interests, authors' contributions and funding in the manuscript.

With these revisions, we hope our paper will be suitable for publication in *Royal Society Open Science*.

Yours Sincerely

Damian J. Ruck

Reviewers comments in ***black italics***, our responses in **red** and text added to the manuscript is in **blue**

Associate Editor's comments (Professor Len Thomas):

Comments to the Author:

Your paper has been read by two reviewers. The first raises a number of issues, including concern about the logic underlying the main premise that secular-rationality preceded WIERDness. They also found your analyses hard to follow from the descriptions in the main paper. The second reviewer was more positive, raising only minor concerns. Given this, I am recommending major revision of the paper. Please address the reviewers' concerns either by revising the paper itself or, if you disagree, by responding fully in your response letter. Regarding the analysis description, I would like to see more detail on this in the main paper, rather than in supplemental materials. In addition to the reviewers' comments, I would also like the following points addressed.

We have expanded the section prior to the regression results, describing the derivation of our time series for Secular-Rationality and Cosmopolitanism, including the empirical justification for using birth decade as our temporal variable. We think this will give the reader a better sense that they have seen the data.

Starting with the raw World and European values survey (WEVS), we used two exploratory methods in sequence to reduce the 68 common WEVS questions to two orthogonal multi-variate components. First, we use Exploratory Factor Analysis (EFA) to identify nine cultural factors underlying the 476,583 question responses. We then interpreted each of these cultural factors in terms of a small and unique set of correlated WEVS questions. For example, the factor we label 'Secularism' is highly correlated with WEVS questions such as, "How important is religion in your life?" and "How important is God in your life?" (Supplementary Materials). By using EFA in this first step, we create a summary of only the common variance, which means noise, such as measurement error, is reduced.

In the second step, we ran a principal component analysis (PCA) on the EFA-weighted WEVS data from the previous step, which gave us a reduced orthogonal representation of the common WEVS variance. This weighted PCA procedure combines advantageous features of both EFA and PCA. Using PCA provides a reduced orthogonal representation of the WEVS data, which reduces collinearity issues in subsequent regressions. Using EFA makes the components more interpretable by minimizing noise.

The first two principal components (PC) explain 37% of the common WEVS variance. We retained PC1 and PC2 because, in our subsequent multilevel time-lagged regressions, they both show strong linkages to the various measures of prosperous societies (life expectancy, education, democracy and GDP per capita), whereas PC3 and PC4 do not (Supplementary Materials). We label PC1 as Secular-Rationality and PC2 as Cosmopolitanism based on the EFA factors that they are highly Pearson correlated with ($|r| > 0.4$). See table 1.

Secular-Rationality is correlated with secularism ($r=0.76$), political engagement ($r=0.62$), respect for individual rights ($r=0.59$) and low prosociality ($r=-0.45$). This means that Secular-Rational respondents to the WEVS are those who reported, for example, that religion is important in their lives, that they are likely to attend protests or sign petitions, they only pay taxes when coerced and believe that homosexuality and divorce are justifiable. Cosmopolitanism is correlated with trust in out-groups ($r=0.78$), trust in norm violators ($r=0.78$) and respect for individual rights ($r=0.43$). This means Cosmopolitan individuals report willingness to have neighbors that are foreign, homosexual, of from another race, as well as believing that homosexuality and divorce are justifiable.

One challenge in our use of the WEVS is that it only stretches back to 1990. To study the slow emergence of prosperous societies during the 20th century we extend the time horizon of the WEVS data to 1900 by treating birth decades as representative of historical time periods (Ruck et al. 2018, Foa et al. 2016). This is possible because cultural values are formed and hardened during the first few decades of life, meaning those who came of age during the 1930s, when surveyed today, will have cultural values that reflect that era. This is based on convergent interdisciplinary evidence from

childhood development (Grusec 1997), political belief formation (Jennings1996, Sears1999), prosociality in small-scale societies (House2013a) and neuroscience (Glimcher et al. 2016, Sowell 1999, Petanjek et al. 2011).

To treat birth decade as our time variable, we ran additional tests. First, environmental shocks cause transient changes in cultural values at particular time periods (Bentzen et al. 2018, Gelfand et al. 2011, Bentley et al. 2014, Henrich et al. 2019), which could systematically affect certain birth decades more than others. However, we use model comparison (two-fold cross validation) to show that cultural value differences between birth decades are stable through time (Ruck et al. 2018, Inglehart 2008), which tells us that shocks generally affect the entire population, not individual birth decades. Second, in the absence of survey data from the early 20th century, we use simulations to show that birth decades are representative of past time periods, even in the presence slow time period effects and uncertainty regarding when a birth decade enters the adult population. We show that the simulation results and the subsequent time-lagged regression results are robust regardless of whether this age is assumed to be 0-10, 10-20 or 20-30 years (Supplementary Materials).

The regression model appears to be Bayesian; this should be mentioned in the main text, together with an explanation for the priors used. I also wondered why a Bayesian analysis was used here when the previous analyses (PCA, etc) are not Bayesian.

Bayesian regression allows us to put reasonably informative priors on the model parameters, rather than the unrealistic flat priors inherent in the frequentist statistics.

Bayesian analysis was not required in pre-processing (EFA and PCA) because it was performed on the individual-level survey data which had very high N ($5^5 \times 68$), meaning informative priors were not necessary. In addition, I do not know of a convenient Bayesian implementation of exploratory methods such as PCA/EFA.

We add text justifying our use of priors in the methods section and make clear that our regressions are Bayesian throughout the main text.

The weakly informative priors capture the fact that all variables are z-scored, which makes large parameter values very unlikely.

I wondered why three different information-criteria were used for inference? I also note that, unlike what is stated in the text, they give different results. From supp materials Fig 4, DIC and WAIC are lower for model M3 compared with model M2.2, certainly for the Cosmopolitanism value. It's hard to tell exactly how much lower from the figure, but it seems to be >50, which is a lot in IC terms.

We agree that using multiple different information criteria is confusing, particularly given they do not converge and that no error is expressed. We instead use 2-fold cross validation to empirically assess each model by splitting the data into training and testing data. A new model comparison figure is added to the Supplementary Materials and we add the following text to the figure caption in the Supplementary Materials:

b) two-fold cross validation results (performance assessed using 'expected log pointwise predictive density') comparing hierarchical models of increasing complexity that explain cultural value change in terms of continuous birth decade g and categories for time period p and nation n ; where M1 is the simplest model (unique intercept and slope for each nation) and M3 is the most complex (unique intercept and slope for each period-nation combination).

And the following to the main text:

To address this, we use model comparison (two-fold cross validation) to show that cultural value differences between birth decades are stable through time (Ruck et al. 2018, Inglehart 2008), which tells us that shocks generally affect the entire population, not individual birth decades.

Thank-you for providing code and data with the paper in a github repository. Unfortunately, it is currently not in an acceptable state for publication with the paper. Please, at a minimum: (i) add a readme file explaining what the files are and how to run them; (ii) add comments both to the R and python files; (iii) tidy both files up so that they use consistent vertical spacing (the code currently is not organized into sensible “paragraphs” and seems to have random single, double and triple vertical spaces); (iv) put all data files into either the top directory or a data directory that is a subdirectory of the top directory (one is currently in the top directory, another is in a directory one level down and the others are a subdirectory nested 4 below the top directory), (v) ensure that the code contains everything needed to replicate the analyses in the paper *and* the supplemental materials. Please double check whether any more tidying up may be helpful in enabling code reuse.

We have updated the github with code and data making all the analysis reproducible. We have also written an extensive readme file, guiding users through the analysis.

Minor editorial comments:

Page 4: I could not understand the “five different waves p ” in the following sentence “We extended the reach of WEVS surveys, conducted five different waves p over the past 30 years, back to the early 20th century by treating the cultural values in survey responses as representative of the respondent’s birth decade, t (20; 26).” For one thing the p should be enclosed in commas as it is parenthetical. But secondly I’m not sure what these “waves” are. (This is explained in the methods but should be defined where first used.)

We have amended this discussion such that “ p ” and “wave” are no longer included in the main text. This change avoids the “alphabet soup” problem reviewer one was concerned about. We also removed reference to the word ‘wave’.

Page 11, please define the +- notation used after the estimated slopes in the linear regression results. Are these standard errors? If so, perhaps just say this and remove the +- device.

These are standard errors. We have replaced the +- notation throughout.

Page 14, In equation 1, should the β_j in the second formula from bottom be β_{ij} ? Also, unless I missed it, J is not defined.

Yes, that’s right. We have corrected this typo. Now j is defined along with the other model parameters below equation 1.

where, $\beta_{i,j}$ is the effect size of independent variable j in matrix $X_{t-g,j}$ on dependent variable i in matrix $Y_{t,i}$

Reviewers' Comments to Author:

Reviewer: 1

Comments to the Author(s)

This paper measures economic growth and WEIRD values over time, finding that secular-rationalism and cosmopolitanism precede economic growth, but not vice versa. The authors conclude that these value are precedents of WEIRD culture. I was intrigued by the analysis and the questions being asked, but I see two major issues: 1. I'm not sure I think the conclusion is warranted. As I outline in more detail in Point #1 below, I think Henrich and colleagues would say these values ARE WEIRDness, not pre-requisites. If I'm correct in this, then the paper needs to be entirely re-framed.

Addressed in Major point 1 below.

2. *I was confused throughout the paper about the data and the analysis (points 5, 6, 7, 8 below). I walked away mostly understanding what the conclusion was but without a clear sense of having "seen" the data. I think it is possible to give readers a better sense of the data. For example, the Grossmann and Varnum 2015 paper walks readers through the process of testing time series data quite clearly. They give graphs that explain what's going on. Although Grossmann and Varnum used complicated methods, I walked away having a sense that I had "seen" the data. For this paper, I really only feel like I have the conclusion the authors. I can't exactly say how to fix this, but I think it's worth time on the authors' part to try to improve.*

On reflection we agree with this (as did the editor). In response, we added the extended description (see blue text on pages 2-3 of this response, above) on the data pre-processing, prior to presenting the regression results.

Major Points

1. *I'm not sure what it means to say that secular-rationality preceded WEIRDness. Isn't secular-rationality WEIRDness itself? Yes, the WEIRD acronym refers to things like education and economic development, but Henrich and colleagues describe these as *correlates* of cultural WEIRDness. What they're primarily interested in is the WEIRD cultural traits of things like trust and analytic thought. Thus, I worry the paper has the logic backwards here. I'm open to being convinced otherwise.*

We are not referring the psychological traits found in WEIRD societies, but rather to the WEIRD societies themselves (Western, Educated, Industrial, Rich and Democratic). Ultimately, we think the WEIRD acronym is a neutral-sounding encapsulation of a certain type of society found mainly in the West.

That said, the confusion is understandable. Therefore, we no longer refer to WEIRD societies in the paper. We instead refer to "socioeconomic development", which is the process that leads to long lived, educated, democratic and rich societies.

Though this does not change the discussion too much, certain text and references have been removed/added. This includes the title and abstract.

In the centuries since the Enlightenment, the world has seen an increase in socioeconomic development, measured as increased life expectancy, education, economic development and democracy. While the co-occurrence of these features among nations is well documented, little is known about their origins or co-evolution. Here we compare this growth of prosperity in nations to the historical record of cultural values in the 20th century, derived from global survey data. We find that two cultural factors, Secular-Rationality and Cosmopolitanism, predict future increases in GDP per capita, democratization and secondary education enrollment. The converse is not true, however, which indicates that Secular-Rationality and Cosmopolitanism are among the preconditions for socioeconomic development to emerge.

2. For a paper that takes the idea of cultural change so seriously, I was surprised to see the following speculative claim of happenstance, couched in such certain terms: "Through an accident of history, Secular-Rationality, Cosmopolitanism arose as founding principles of western European nations..." Huh? How do we know that's a happenstance and not a change with a meaningful cause? It seems far too speculative to claim this.

We agree that this claim is too strong, and we have replaced it with a more speculative one. Though our paper does not speak to the drivers of cultural change, there is a possibility that Secular-Rationality and Cosmopolitanism were innovations by chance, first occurring in Western Europe, then diffusing to other nations. We've added the following text:

Precisely why Secular-Rationality and Cosmopolitanism arose in Western Europe is not known, it could be due historic institutional innovations (Acemoglu et al. 2019, Schulz et al. 2019) or possibly just an accident of history. Nonetheless, once established these values appear to have diffused through geographic proximity—the Uralic speaking nations of Europe have high Secular-Rationality and Cosmopolitanism—and diffused even more readily into nations with similar cultures and languages (Matthews 2016a, Spolaore et al. 2013, Norris 2009), giving rise to the current observed pattern of national differences in cultural values (Figure 3).

3. The paper mentions the limitation of needing to look at migration but doesn't seem to do much about it. It seems to me like two strategies are available: (1) statistically model movement between nations (following the Putterman and Weil paper) and (2) analyze heritage/mobility/birthplace data in the WVS itself (if it exists).

The reviewer suggests two approaches that could help us control for unknown migration between nations of adults that have already been culturally socialized. Unfortunately, these cannot be applied in our analysis. The Putterman and Weil (PW) dataset does not include data for birth decade (or age) of migrants. Without this, we cannot make informed corrections for the cultural values of birth decades because we do not know which birth decade to correct. In addition, the PW is measured at 50-year intervals which offers a too coarse description of migration during the 20th century.

The reviewer also suggests linking immigration status to age by using the WEVS survey data directly. Unfortunately, though a question exists asking "country of birth", it only appears in 2/5 of the World Values Survey and 0/4 of the European Values Surveys. Therefore, this correction cannot be made for anywhere near the entire WEVS corpus. We add the following text to explain this point:

Future work should identify alternative measures of long-term cultural values because the birth decade differences we measure could be distorted by the migration of already culturally socialized individuals (Kerr et al. 2017). Standard approaches cannot help us correct for this. For example, the use of Putterman and Weil's (2010) migration data are not applicable in our case because they do not track the birth decade of migrants. This may not be a concern because, migrants do tend to converge on the culture of their new society (Mesoudi 2016, Norris 2012, Richerson et al. 2005), meaning Secular-Rationality and Cosmopolitanism are still likely preconditions for the emergence of prosperous nations.

4. Page 2 says, "Long-run time series for socio-economic development variables stretch back to the 19th Century..." I see the authors cite the Maddison project, so perhaps they know this already, but the project includes data (for certain countries) far earlier than the 19th century.

We are aware. We phrased it like this because we are only interested in data since the end of the 19th century. However, we acknowledge that this is confusing, so rephrase it in the following way:

Long-run time series for socioeconomic development variables stretch back to the start of the 20th century and, in some cases, many centuries before that.

5. Page 3 lists what appear to be either correlations or components of the scales, but it's unclear to me: "is secular ($r = 0.76$), politically engaged ($r = 0.62$)..." Please make this clear. The original scale wordings (at least a few) would give readers a lot better idea of what they're reading.

Good point. In our expanded description of the data pre-processing (see blue text on pages 2-3 in this response letter, above), we now explain that "r" means Pearson correlation and tie our cultural factors to the original survey questions. This gives a better view of the data that comprise "Cosmopolitanism" and "Secular-Rationalism".

6. Page 4 describes the use of birth decade and gives the impression the authors did commendable work demonstrating that this is a robust way of analyzing the data. It sounded impressive to me, but I didn't really understand it! One of my biggest questions was, did you try modeling it other ways? For example, what about the possibility that people's values are formed mostly from age 10-20? Perhaps this is already in the analysis. But what I'm asking is the authors to make these statements clear enough that a casual (but educated) reader can understand concretely what the analyses were. Sometimes communicating analyses clearly is harder than doing the analyses themselves! UPDATE: On page 5, the paper seems to address this question. That's great! But the fact that it's addressed on page 5 makes me even more confused about what I had read on page 4. What did I read?? LATER UPDATE: The methods say "However convergent interdisciplinary evidence (6; 27; 28; 29) suggests that decade of birth can be used to represent historical time dynamics that link to socioeconomic development (20; 32)." That's great that there's evidence to support this! But can you describe some of it? Even just a few sentences communicating what those studies have compared to test this idea would be helpful.

We agree, it is important for the description of the analysis to be clear. To this end, we take the reviewers advice and we rearrange and have added additional description – see the paragraph in blue text, above on page 3 in this response letter, beginning with "To treat birth decade as our time variable..." We also move the "convergent interdisciplinary evidence (6; 27; 28; 29)" to page four and expand on these studies: see the blue text on page 2, above, beginning with "This is possible because cultural values are formed and hardened during the first few decades of life..."

7. Page 4 describes the analysis using an alphabet soup that is cognitively taxing for the readers: "changes in C and R...past L on future D." Yes, these are defined elsewhere, so it's possible for readers to figure it out. But why make their job so hard? It adds too much complexity for the readers to try to keep all these symbols in mind. Replacing a single letter with a word or two (like "democracy") will be well worth the extra space taken up. It's far worth the cost in order to have readers be able to actually understand the paper.

We agree with this point, retaining these abbreviations creates unnecessary cognitive work for the reader. Therefore, all in-text references to variables are now the full capitalized name (e.g. D = Democracy).

8. Page 6 mentions "Uralic" languages. My guess is even most educated readers won't know what that means. It would be helpful to describe that (even just parenthetically) in the paper. (Ditto for Kartvelian!)

In line with minor point 5, we relabel "Kartvelian" as "Georgian". We state the European nations that speak "Uralic" languages.

High S_{R+C} nations are found in Germanic, Japanese and Uralic (spoken in Hungary, Estonia and Finland) language families. Then with lower, but still positive, S_{R+C} in Italic, Balto-Slavic, Korean and Sino-Tibetan families. On the other hand, the negative S_{R+C} language families are Albanian, Austroasiatic, Tai, Indo-Aryan, Turkic, Semitic, Niger-Congo, Austronesian and Georgian.

Minor Points

1. For this sentence, I think readers would appreciate a basic correlation to get an idea of the relationship. "These two components have strong linkages to measures of socioeconomic development (Supplementary Materials)."

The strong linkages are statistically demonstrated by the analysis, so we have removed this sentence. As detailed in blue text on page 2, above, we now explain why PC3 and PC4 were not retained (they do not show strong linkages to the various measures of prosperous societies).

2. The paper says the analysis has a "credible interval of 95%." Is that another term for confidence interval? I've never heard this term before.

Credible interval is the Bayesian analogue to the frequentist confidence interval. Because we are in the Bayesian framework, we can assign probability distributions to our parameters and so we report whether 95% of the probability distribution excludes zero (analogous to a 5% p -value). We add the following clarifying sentence:

As we are using a Bayesian model, we report expected effect sizes and whether 95% credible interval of the posterior distribution excludes zero.

3. Page 10: "how that" sounds like a typo to me.

Corrected.

4. Page 8: "the later" should be "the latter."

Corrected

5. Japanese and Kartvelian are described as language families, yet both of these represent a single language. In those cases, the wording in the paper will mislead readers who don't already know this (somewhat obscure) fact.

We agree. We rename "Kartvelian" as Georgian and add the following sentence that acknowledges single language families:

To test this, Figure 3b plots the average S_{R+C} for each language family in our data, this includes the following single language families: Japanese, Korean, Austroasiatic, Tai and Georgian.

Reviewer: 2

Comments to the Author(s)

This is a very interesting paper. In my opinion, it only needs some minor changes. I appreciate that the authors put their data and code on github. But as it is right now, there is no sort of README file or guidance on how to reproduce their data. To replicate, I would simply try to run the two source codes they provide, but I would really appreciate some file telling me what to run to produce what figures.

In accordance with the Editor's request, the github repository has been updated and a detailed readme file added.

One bit that concerned me is the use of language similarity as a proxy for cultural proximity to western Europe. To me, this is concerning because while many former colonies do share the language of their colonisers, it does not imply at all that they share their values at the time, or even today. While the authors do point that out on page 11 ("This is reasonable because migrating Europeans carried their culture with them, which does not necessarily mean transplanting historic cultural values of the day, but just more stable cultural features like language"), I really do not see why sharing a language would imply sharing values.

This is an excellent point and we do not want to be misunderstood. We have edited this section to make our case clearer, covering the following points:

- We see cultural values as the "software of society" that can be innovated in one society and adopted by another.
- Language is a reasonable cultural proxy because of culture is often transmitted linguistically. This means there is a greater *potential* for culture to be transmitted between linguistically similar societies, though it is not guaranteed.
- We have shown empirically that linguistic distance from Germanic speaking nations in Western Europe predicts levels of Secular-Rational + Cosmopolitan values. We simply state that this could be due to the spread of these cultural values along linguistic (or cultural) lines, which the pattern does suggest.

To take the English language as an example: the UK, Belize, Guyana, Jamaica and the USA speak English. And as shown in the Supplementary Information, the Germanic family includes Austria, Germany and the UK, but also Singapore, Zambia, Ghana and Nigeria. Would they all have a regression term corresponding to the English language? Perhaps I understood this wrong, but this sounds concerning to me. Perhaps this is addressed by the references (35; 36; 43) cited in the end of the first paragraph of page 11, but as the references are not numbered I am not sure which reference is which. In any case: I think the authors should discuss this assumption -- and this classification of such a wide variety of countries into "Germanic countries", for instance -- a bit further.

First thing, we apologize that the references were not numbered. We have corrected this. To the more substantive point, you are right to say that there are exceptions to the rule. One example from our data is Trinidad and Tobago, which is linguistically classified as English, but has an S+C value more in line with Spanish speaking Latin America. However, the former African colonies of European nations (such as Nigeria) are classified using the native African language family, "Niger-Congo". This is important because English speaking is not universal in these nations and an array of Niger-Congo languages are still spoken by millions of people. We felt, therefore, it was important to distinguish these nations from the cases like the United States and Australia, where English speaking is

universal. Nonetheless, this is an understandable point, so we've clarified this section as follows:

The large global variation in Secular-Rationality and Cosmopolitanism is related to the linguistic and cultural history of nations (Inglehart 2000, Basanez 2016a). To quantify this, we make a simple cultural metric that is the sum of the Secular-Rationality and Cosmopolitanism scores, S_{R+C} . Figure 3(a) illustrates the distribution of S_{R+C} on a world map. The regions with the highest S_{R+C} are in Western Europe and their historic colonies in Australasia and the Americas. From inspection, this map suggests that cultural distance from Western Europe in part determines S_{R+C} .

Cultural values can be viewed as the "software of society" (Hofstede et al. 2010). We hypothesize that cultural values can be innovated in one place and spread to another. Evidence suggests that Secular-Rationality and Cosmopolitanism were likely innovated in post-Enlightenment Western Europe (Pinker 2018, Mokyr et al. 2016). Under our hypothesis, these cultural values can then diffuse from one nation's population to another, which occurs more readily between nations that are geographically close or linguistically similar because barriers to communication are lower (Matthews 2016a, Spolaore et al. 2013, Norris 2009). Therefore, we expect to see high S_{R+C} in nations that are either linguistically similar or geographically proximate to the Germanic speaking nations in Western Europe.

To test this, Figure 3b plots the average S_{R+C} for each language family in our data, this includes the following single language families: Japanese, Korean, Austroasiatic, Tai and Georgian. We classify former-colonial African nations (such as Nigeria) using the Niger-Congo language family, despite many people in the region speaking European languages. We do not classify these nations using European languages because millions of people still speak native languages (full classifications in Supplementary Materials).

As expected, Figure 3b shows that European language families are associated with relatively high S_{R+C} . High S_{R+C} nations are found in Germanic, Japanese and Uralic (spoken in Hungary, Estonia and Finland) language families. Then with lower, but still positive, S_{R+C} in Italic, Balto-Slavic, Korean and Sino-Tibetan families. On the other hand, the negative S_{R+C} language families are Albanian, Austroasiatic, Tai, Indo-Aryan, Turkic, Semitic, Niger-Congo, Austronesian and Georgian.

Language families are discrete categories, so we use ancestry information as a continuous proxy for cultural proximity to Western Europe. We assume that cultural distance from European nations is correlated with the proportion of modern day populations descended from historic European populations in 1500 (Putterman 2010). This is reasonable even if Secular-Rational and Cosmopolitan cultural values are not transmitted directly because migrating Europeans also carry more stable cultural features such as language and religion. Having a similar language and religion reduces barriers between Europe and the new nation, which means Secular-Rational and Cosmopolitan values can diffuse more easily once innovated (Matthews 2016a, Spolaore et al. 2013, Norris 2009).

We fit a linear regression to predict the effect of European ancestry on S_{R+C} , testing for the influence of the three European linguistic families with large sample sizes ($n > 10$) --- Germanic, Italic and Balto-Slavic. As Figure 3c shows, Germanic ancestry has the largest effect (slope = 1.53, standard error 0.11 and $p \approx 0$), Italic has a smaller effect (slope = 0.9, with standard error 0.12 and $p \approx 0$) and Balto-Slavic has the smallest effect (slope = 0.58, with standard error 0.1 and $p \approx 0$). Italic languages are culturally closer to Germanic than Balto-Slavic, suggesting cultural distance from Germanic speaking nations is historically important.

Other than that: on page 9, the authors say "Uralic speaking countries of Europe exhibit high S_{R+C} ". Would that be S_{R+C} ?

Yes, quite right. Corrected.